# Correlation Between Stoichiometry of $Nb_xN_y$ Coatings Produced by DC Magnetron Sputtering with Electrical Conductivity and the Hall Coefficient

**Angélica Garzon-Fontecha [1], Harvi A. Castillo [2], Daniel Escobar-Rincón [3]** 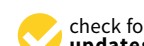**, Elisabeth Restrepo-Parra [3],* and Wencel de la Cruz [2]**

[1] Centro de Investigación Científica y de Educación Superior de Ensenada (CICESE), Carretera Tijuana-Ensenada No. 3918, A. Postal 360, 22860 Ensenada B.C., Mexico; agarzon@cicese.edu.mx

[2] Centro de Nanociencias y Nanotecnología, Universidad Nacional Autónoma de México, km. 107 Carretera Tijuana-Ensenada, C.P. 22860 Ensenada B.C., Mexico; hcastillo@cnyn.unam.mx (H.A.C.); wencel@cnyn.unam.mx (W.d.l.C.)

[3] Laboratorio de Física del Plasma, Universidad Nacional de Colombia Sede Manizales, Manizales 170003, Colombia; danielescobarj@gmail.com

**\*** Correspondence: erestrepopa@unal.edu.co; Tel.: +57-6-887-9495

**Abstract:** Non-stoichiometric $Nb_xN_y$ coatings, produced in a reactive sputtering process, were analyzed on the basis of their chemical composition (specifically, nitrogen concentration) and its relationship with electrical conductivity. The chemical composition and bonding configuration were examined using X-ray photoelectron spectroscopy (XPS), revealing Nb–N bonds. The stoichiometry variation dependence on the $N_2$ flow was also analyzed, using Auger electron spectroscopy (AES). Without exposing the samples to air, a normal behavior was observed; meaning that the nitrogen concentration in the coatings increased, with an increase in $N_2$ flow. The electrical properties were evaluated and their relationship with nitrogen content in the films was analyzed. The highest conductivity value for all studied samples was observed for the sub-stoichiometric film, $NbN_{0.32}$, which also exhibited a positive Hall coefficient. It indicated that the conduction was mainly dominated by hole-type carriers. High conductivity at lower nitrogen content was attributed to the fact that, at a low concentration of nitrogen, the effect of impurities, acting as dispersion points for electrons, was lower, increasing the relaxation time. As the main conclusion, the $Ar/N_2$ flow ratio strongly influenced the coatings of stoichiometry and then, this stoichiometry affected, to a great extent, the electrical conduction and the Hall coefficient of the coatings.

**Keywords:** nitrogen flow; XPS; Auger electron spectroscopy; four points; Van der Pauw method

## 1. Introduction

Transition metal nitrides and carbides have the potential to serve as highly corrosive-resistant materials, barrier layers for semiconductor materials [1], and hard coatings [2]. Many of these ceramic materials exhibit great thermal and chemical stability, high resistance to wear, corrosion and oxidation, being attractive for applications in the manufacturing industry [3]. Niobium nitride, which is a member of hard nitride coatings family, was initially investigated for its superconducting properties, due to its high critical current density, excellent mechanical properties, and high superconducting transition temperature (in the order of 17 K) [4]. On the other hand, NbN has been produced as monolayers, using techniques such as chemical vapor deposition [5], atomic layer deposition [6], magnetron sputtering [7–9], cathodic arc process [10], sol–gel [11], plasma immersion ion implantation [12], among others. According to Zhitomirsky [13], the complicated Nb–N phase diagram [14] and the

strong dependence of Nb–N crystalline structure and texture on deposition parameters, such as nitrogen pressure [15], substrate bias voltage [16], and even layer thickness [17], lead to different, often contradictory and even confusing, data on binary NbN and NbN-based multi-component and multi-layer coatings. For instance, there have been some studies dedicated to studying the influence of $N_2$/Ar flow ratio on the mechanical and tribological properties of NbN produced by magnetron sputtering [18]. Du et al. [19], presented studies on the preparation, structure and optical properties of NbN films deposited by DC reactive magnetron sputtering, at different $N_2$ partial pressures and different substrate temperatures, ranging from −50 to 600 °C. With the increase in the $N_2$ partial pressure, a δ-NbN phase structure is formed and the grain size and lattice constant of the cubic NbN increases. Singh et al. [20] presented a study of NbN films deposited on MS, SS, and HSS substrates, by a reactive DC magnetron sputtering, studying the effects of $N_2$ flow and substrate biasing on the deposition rate, crystal structure, surface hardness, adhesion, and tribology.

Several studies have been made on the conductivity of NbN films. Menon et al. [21] studied the influence of partial nitrogen pressure on the conductivity of NbN, at temperatures near the superconducting point, where a wide range of electrical and superconducting properties varied with the $N_2$ flow. Nigro et al. [22] and Polakovic et al. [23] related the microstructural parameters (grain boundaries) of the films with conductivity in a wide range of temperatures (101–102 K), and found that the charge transport is strongly influenced by grain boundary scattering of the conducting electrons. Finally, Cabanel et al. [24], reported the initial formation of NbN films and thereafter oxidation, due to air exposure of the material and the important role of the oxidated phase ($NbN_xO_y$), which establishes energy barriers for electrons between grains. Although previously mentioned works have focused on NbN conductivity and its relationship with material features and synthesis parameters, none of them have explored, in detail, the stoichiometry in a wide range of $N_2$ content and the final charge transport properties. Specifically, for NbN coatings, up-to-now, there is lack of experimental data regarding the effect of deposition parameters, as $N_2$/Ar flow, on the coating stoichiometry and especially, on the electrical behavior far from low transition temperatures; therefore, it is important to consider that nowadays, great efforts are being carried out for extending NbN applications, such as highly sensitive thermometers for ranges close to RT and their potential use in nanomagnetism, biophysics, and infrared bolometry [25].

The aim of this work was to produce NbN coatings, using the magnetron sputtering technique and to study the relationship between the $N_2$/Ar flow ratio and stoichiometry of the synthesized materials. These results were conducted to enlighten the influence of $Nb_xN_y$ stoichiometry, and the consequent amount of Nb–N bonds, on the electrical properties, at a temperature range of 100 to 350 K.

## 2. Materials and Methods

Coatings were synthetized using the DC magnetron sputtering technique on silicon (111) and glass substrates, employing a target of Nb. Films grown on glass substrates were used for measuring the electrical properties. The Nb target was 5 cm in diameter and kept at 8 cm from the substrate. Before the coatings production, a vacuum pressure of the order of $10^{-5}$ Torr was reached and the reaction chamber was baked, performing enough degassing over 12 h. using a heat tape, at a temperature of 200 °C. After this procedure was finished, the chamber was cooled for one day, until reaching the room temperature (RT). Coatings were produced at a DC power of 50 W and with a mixture of Ar and $N_2$. Deposition pressure for growing the samples was 8 mTorr. All obtained coatings remained at a thickness of around 350 ± 20 nm, during 1200 s, in order to perform a suitable comparison. For this purpose, thicknesses of some chosen samples, deposited at each nitrogen flow and for certain time periods, were measured by the atomic force microscopy (AFM XE-70 from Park Systems in contact mode, Santa Clara, CA, USA) and deposition rates were calculated, in order to extrapolate the required time to achieve the desired thickness. Although the $N_2$/Ar ratio was varied, the thickness did not exhibit changes above values of 20 nm. The experiments were performed with argon flow of 10 sccm, varying the nitrogen flow between 0 and 2.5 sccm. After that, coatings were moved to

a vacuum chamber ($10^{-9}$ Torr) for XPS and AES in situ analyses. The XPS and AES analyses were obtained (PHI model 548 spectrometer manufactured by Physical electronics, Chanhassen, MN, USA). The spectra were acquired using kinetics energies between 20 and 1000 eV (with steps of 1 eV) and the information was taken at one scan per min. The XPS spectra were taken at a high resolution, using Al (K$\alpha$) X-ray with energies of 1486.6 eV. The high resolution spectra were taken with an energy of 50 eV and a step of 0.2 eV for Nb 3$d$, N 1$s$, and O 1$s$. The factor of sensibility used for calculating the elemental concentration, according to Scofield were 1.8, 2.93, and 8.21 for N, O, and Nb, respectively [26]. In order to determine the conductivity and Hall coefficient of samples, the Van der Pauw method was applied, using HMS-5000 from Ecopia (manufactured by Bridge Technology, Chandler, AZ, USA). The contact electrodes consisted of indium–tin and measurements were acquired by varying the temperature between 85 and 350 K.

## 3. Results and Discussion

After coatings production, X-ray diffraction patterns were measured for all $Nb_xN_y$ samples. Characteristic sharp peaks corresponding to crystalline phases were not found and instead, a broad intensity, commonly attributed to a disordered atomic distribution, were observed. Amorphous transition metal nitrides have been extensively produced by the reactive sputtering technique [27,28], and this resulted in structures that have been attributed mainly to the reduced atom mobility on the substrate surface, due to its temperature (room temperature) and due to the inequality of Nb and N atoms that arrive onto the substrate, depending on the nitrogen flow, and hence, the incorporation of atoms in the final formed materials [29].

### 3.1. AES and XPS Characterization

Figure 1 shows the sequence of Auger spectra obtained for the $Nb_xN_y$ films grown on silicon (111) substrate. A main signal placed at 167 eV, corresponding to the Nb LLM, was observed. The spectrum (Figure 1a) corresponds to the metal Nb. In the case of the spectra between (Figure 1b) and (Figure 1e), a peak of N KLL, placed at 379 eV, was identified.

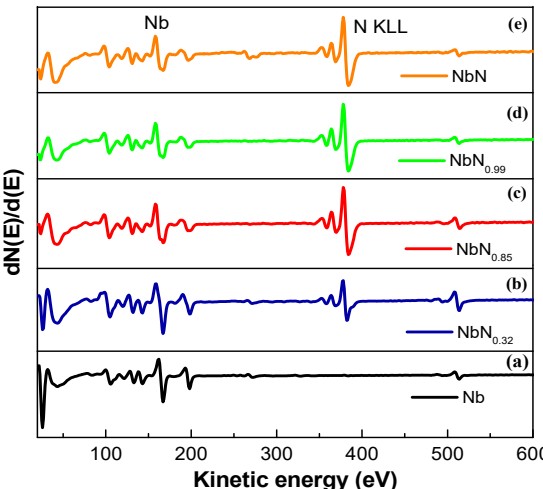

**Figure 1.** Auger electron spectra (AES) for $Nb_xN_y$ coatings, by varying the nitrogen concentration: (**a**) Nb, (**b**) $NbN_{0.32}$, (**c**) $NbN_{0.85}$, (**d**) $NbN_{0.99}$, and (**e**) NbN.

Additionally, a peak at 503 eV was observed, due to the oxygen impurities, since residual oxygen gas was not completely evacuated and included in the films, in the deposition process. Table 1 includes the percentage composition and stoichiometry of the $Nb_xN_y$ layers, obtained by Auger electron spectroscopy, by varying the $N_2$ flow between 1 and 2.5 sccm. According to the Auger results, NbN

stoichiometric coatings were obtained using a mixture of Ar and $N_2$ flows of 10 and 2.5, respectively. The quantification was performed according to the methodology explained in the literature [19,20].

**Table 1.** Stoichiometry of $Nb_xN_y$ thin films calculated using AES, by varying the nitrogen flow between 1 and 2.5 sccm; Ar gas flow was constant (10 scccm).

| $N_2$/Ar Ratio | $N_2$ Flow (sccm) | Nb (%) | N (%) | $Nb_xN_y$ |
|---|---|---|---|---|
| 1/10 | 1.0 | $75.5 \pm 1.4$ | $24.5 \pm 0.9$ | $NbN_{0.32}$ |
| 1.5/10 | 1.5 | $54.0 \pm 1.5$ | $46.0 \pm 0.8$ | $NbN_{0.85}$ |
| 2/10 | 2.0 | $50.3 \pm 1.8$ | $49.7 \pm 0.9$ | $NbN_{0.99}$ |
| 2.5/10 | 2.5 | $50.0 \pm 1.5$ | $50.0 \pm 0.9$ | $NbN$ |

As the nitrogen flow reached higher values, an increment in the nitrogen concentration was also observed, obtaining compositions up to stoichiometric NbN. This dependence was expected because at a low nitrogen flow, there is a deficiency of nitrogen to interact with niobium, due to the small amount of N atoms in the plasma that can react and reach the substrate surface, resulting in a low concentration of this element in the nitride compound. As $N_2$ flow increases, metal and gas interaction cross-section (and their capacity to react) increases and elemental percentages tends to be similar to the stoichiometric composition [30,31]. For the present research work, it is highly important to determine the nitrogen content with high accuracy, since the N content has a strong influence on the resultant disordered structure of the nitrides and in their final electrical properties. AES has been found to have a high interaction cross-section in the case of low atomic number elements, which enhances the signal peak for nitrogen; on the contrary, XPS exhibits a greater cross-section for high atomic number elements, such as Nb [32]. According to this, stoichiometries reported in the present work are obtained through AES. Moreover, although both XPS and AES can give information about the elements concentration and both of them allow the determination of the molecular structure, AES shows a better spatial resolution than XPS, allowing the identification of small particles, since electrons can be focused in a small area, smaller than X-rays. Nevertheless, the higher energy of incoming electrons in AES generates a larger potential for surface damage, than does the XPS.

Alongside Auger analysis, chemical composition of the $Nb_xN_y$ thin films was also studied using the XPS technique. High-resolution spectra of Nb 3*d* and N 1*s* were taken and are presented in Figure 2a,b, respectively, for several $N_2$ concentrations. In the case of Nb 3*d*, a decrease in the peak intensity was observed and the binding energy (BE) decreased with the $N_2$ flow increment. On the contrary, the intensity of the N 1*s* peak increased, as the nitrogen flow increased.

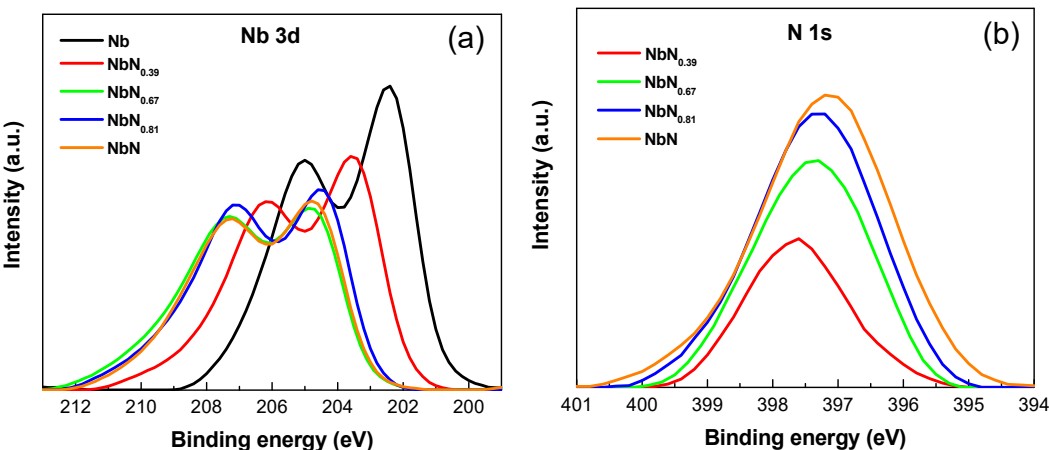

**Figure 2.** High resolution X-ray photoelectron spectra of the $Nb_xN_y$ thin films by varying the nitrogen concentration: (**a**) Nb 3*d* and (**b**) N 1*s*.

From these compositional results, coating stoichiometries were calculated for the $Nb_xN_y$ samples and the values are summarized in Table 2. As is observed in these results, as the $N_2$ flow increased in the coatings production, the percentage of nitrogen in the $Nb_xN_y$ also increased, corroborating the Auger results and proving the influence of $N_2$ flow on the final composition of the obtained materials.

**Table 2.** Composition of the $Nb_xN_y$ thin films grown at different nitrogen flow obtained from the XPS results. Ar gas flow was kept constant (10 sccm).

| $N_2$/Ar Ratio | $N_2$ Flow (sccm) | Nb (%) | N (%) | $Nb_xN_y$ |
|---|---|---|---|---|
| 1/10 | 1.0 | $71.8 \pm 1.1$ | $28.2 \pm 2.1$ | $NbN_{0.39}$ |
| 1.5/10 | 1.5 | $59.8 \pm 1.1$ | $40.2 \pm 1.9$ | $NbN_{0.67}$ |
| 2/10 | 2.0 | $54.9 \pm 1.3$ | $45.0 \pm 1.9$ | $NbN_{0.81}$ |
| 2.5/10 | 2.5 | $49.9 \pm 1.1$ | $50.0 \pm 1.3$ | NbN |

These results are clearly shown in Figure 3. This figure shows the binding energy (BE) of Nb 3*d* and N 1*s* peaks, depending on the N/Nb ratio for the films of $Nb_xN_y$ grown by varying the nitrogen flow, and a shift was also observed for the BE of both elements (Nb 3*d* and N 1*s*). Differences between stoichiometries obtained by XPS and AES can be attributed to the XPS, which can cover a greater surface area than the AES. The AES can give results of small areas with a localized spot.

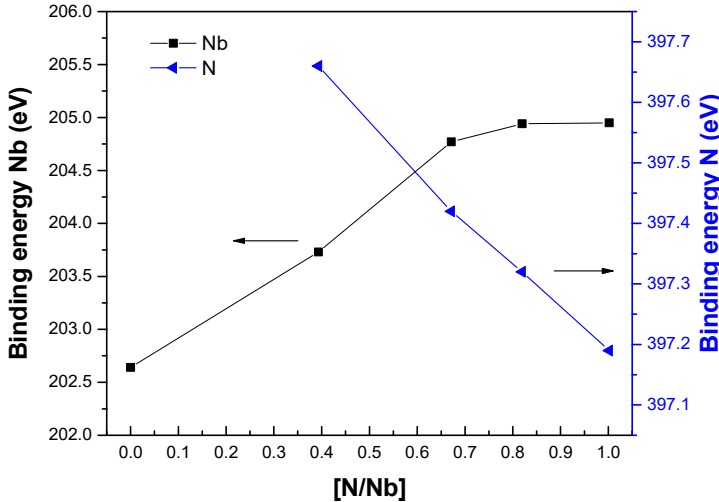

**Figure 3.** Binding energy of Nb 3*d* 5/2 and N 1*s* of $Nb_xN_y$ thin films, as a function of the nitrogen concentration.

Additional to elemental analysis, XPS can also provide information regarding the charge transfer of conducting materials, related to the binding energy shifts of metallic and non-metallic elements of the nitride. The shifting of Nb 3*d* peak was observed to be in the range of ~202.5 and ~205.0 eV; while peaks of N 1*s* varied between 397.2 and 397.6 eV. It was evident that the increase in binding energy of the Nb 3*d*, and in contrast, the decrease of the binding energy of the N 1*s*. This shifting was mainly attributed to changes in the stoichiometry as the nitrogen concentration was increased. It is important to note that, Nb $3d_{5/2}$ and N 1*s* binding energy difference (apparent binding energy or ABE) was related to a ligand charge transfer [33], and a higher difference (~194 $\pm$ 0.04 eV) was found for the sample grown at 1.0 sccm of nitrogen flow, allowing to conclude that a charge transfer for this sample was greater than that in the other cases. For other samples, as ABE of Nb $3d_{5/2}$ and N 1*s* became lower (~192 $\pm$ 0.04 eV), the charge transfer decreased, as was found later in the electrical conductivity analysis.

### 3.2. Electrical Conductivity of Coatings

Normally, electrical properties of the $Nb_xN_y$ have been studied around the superconductivity region; in our work, we decided to study the region around room temperature, that allows many important technological applications, such as highly sensitive thermometers for ranges close to RT and is also potentially useful for nanomagnetism, biophysics, and infrared bolometry [25]. Figure 4 shows the conductivity as a function of temperature for three samples Nb, $NbN_{0.32}$, and NbN. Different behaviors were observed, demonstrating that the addition of nitrogen strongly affected the electrical properties. Beginning with the curve for pure Nb, it was observed that a decrease in conductivity depended on the temperature.

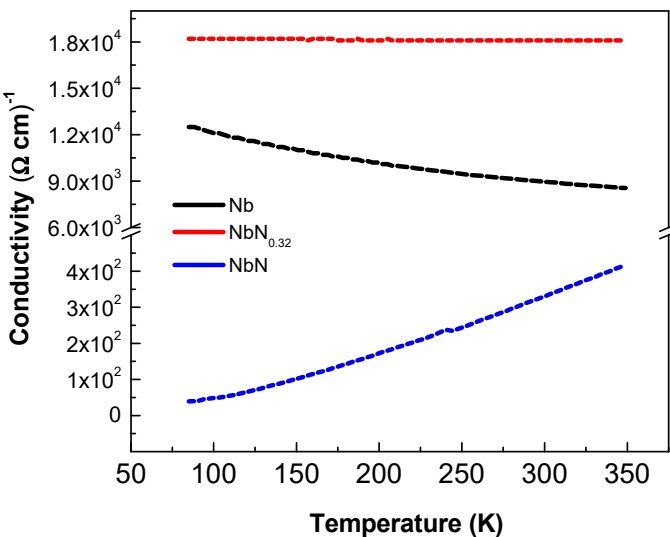

**Figure 4.** Electrical conductivity of Nb, $NbN_{0.32}$, and NbN as a function of temperature.

This is a typical behavior of the metal materials, produced by the increase of vibration amplitudes of ions in the metal, making it more difficult for the electron movement, increasing the electron–ion collisions. These collisions avoid the electrons drift, decreasing the electrical current. Over a small temperature range, the resistivity of a metal can be represented by a linear relation; for instance, in the case of pure Nb. Cannelli et al. [34] reported a decrease in the conductivity as the temperature increased and can be represented as two different polynomials, depending on the region of the temperature. On the other hand, for the case of low concentrations of nitrogen, such as $NbN_{0.32}$, the influence of the temperature on the conductivity was almost neglected. Since the low concentration of nitrogen entailed a disorder in the sample, the lower-dependence of conductivity on the temperature, might be due to impurity scattering, which is not affected by temperature. In this scenario, the contribution of electron–phonon scattering, which increases with the temperature, is negligible, compared to the impurity scattering [35]; then, the mobility of the free carriers is mainly dominated by the vacancies and impurities. A similar behavior was reported by Sugimoto and Motohiro [1]. They reported an insignificant increase in resistivity, for low nitrogen concentrations, as the temperature decreased; nevertheless, they did not give a suitable explanation, since they were focused on the superconducting transitions at low temperature. For the case of stoichiometric NbN, the conductivity tended to increase (resistivity tended to decrease) as the temperature increased. Non-metallic conduction, which is often observed in highly nanocrystalline and amorphous microstructures, is ascribed to the great quantity of point defect and scattering effect in these distortions [22,34,35]. Part of these results is shown in Table 3, where the conductivities of Nb, $NbN_{0.32}$, and NbN at room temperature (RT) were included. RT is taken at a temperature of ~300 K.

**Table 3.** Electrical properties of some $Nb_xN_y$ thin films.

| Sample | $N_2$ Flow (sccm) | Conductivity $(\Omega\,cm)^{-1}$ | Hall Coefficient $(cm^3/C)$ |
|---|---|---|---|
| Nb | 0.0 | $8.90 \times 10^3$ | $-2.18 \times 10^{-4}$ |
| $NbN_{0.32}$ | 1.0 | $1.81 \times 10^4$ | $4.00 \times 10^{-5}$ |
| NbN | 2.0 | $3.33 \times 10^2$ | $-2.14 \times 10^{-4}$ |

These values corroborated the conductivity tendency regarding the nitrogen inclusion in the Nb sample. These values were lower than those reported for bulk Nb ($6.57 \times 106\ (\Omega\,cm)^{-1}$). This behavior is due to thin films presenting a greater quantity of voids (including at the borders of the surface), than the bulk material.

### 3.3. Hall Effect Analysis

Table 3 also includes the Hall coefficient ($R_H$) for Nb, $NbN_{0.32}$, and NbN thin films that showed values different to the Hall coefficient for bulk Nb, which was reported to be $9 \times 10^{-5}\ cm^{-3}/C$ [36,37]. These differences are attributed to the greater number of voids and the loss of long-range order in the atomic distribution (non-crystallinity) of atoms, formed during growth. Moreover, the properties of Nb-based coatings are strongly affected by the processing method; for instance, it was reported that Nb films produced by magnetron sputtering at room temperatures, exhibit a low interdiffusion, being an excellent choice for several technological applications [38–40]. On the other hand, an important observation in our results was the change in the Hall coefficient sign from negative to positive and vice versa. Several authors have reported this change of sign for the case of Nb thin films, based on positive $R_H$ values for bulk Nb; nevertheless, similar to other published studies, $R_H$ was negative for the Nb thin films produced in this study. This change in sign was attributed to several effects, based on the presence of two type of carriers, holes and electrons; both of them contributed to the film electrical conductivity. The contribution of holes normally dominates the electrical transport in thicker coatings, similar to the bulk niobium; on the contrary, the electron contribution strongly dominates in thinner coatings [37], this electron distribution can originate from several effects, such as (i) the Nb band structure in amorphous thin films changes, producing a variation in the carriers type that dominate the conduction; (ii) since the coatings are produced on silicon substrates, an amorphous silicide niobium layer can be formed at the interface, entailing the production of a second conduction channel; and (iii) the scattering process generated at the interfaces and high deformation density zones (consequence of a highly atomic disordered structure) trying to equilibrate the relaxation times of the two carriers types, in different Brilloiun zones, contrary to the case of bulk Nb, where relaxation times for holes are larger than for electrons that are scattered more strongly by impurities [41]. It is important to note that even the atomic ordering of the obtained material has not been observed; it has been shown that disordered materials can present a band structure with reasonably well-defined absorption edges, corresponding to a band gap [42–44]. According to our results, $R_H$ became positive when a small quantity of nitrogen was introduced. This behavior can be attributed to this small quantity of nitrogen atoms acting as impurities, which mainly disperse the electrons, favoring the conduction by the hole-type carries. As the nitrogen level increased, reaching the stoichiometric compound (NbN), the nitrogen atoms were no longer considered to be impurities and the material became similar to the Nb thin film. In this case, the conduction was dominated by the electrons, because of the changes in the band structure, due to the formation of a Si/NbN interface between the substrate and the coating and the great scattering of the electrons generated at the interfaces and the local deformed zones.

## 4. Conclusions

Thin coatings of $Nb_xN_y$, with varying stoichiometry, were grown using a magnetron sputtering system; varying nitrogen/argon ratio and their properties, such as chemical composition, electrical conductivity, and Hall effect were characterized. XPS results showed an increase in the binding energy of Nb, and a decrease in the binding energy of the N 1*s*. These variations were attributed to changes in the stoichiometry and the formation of different phases.

A strong influence of stoichiometry on the temperature dependence of conductivity was also observed. Pure Nb was highly affected by the temperature because of the increase in the phonon vibrations; low nitrogen impurities concentration materials were almost unaffected by the temperature, as the electron scattering was mainly due to the impurities. Finally, in stoichiometric NbN, the conductivity tended to increase. This behavior was ascribed to a short-range order of atomic distribution and defect scattering effects, which is a typical characteristic of amorphous materials. The Hall coefficient in all films showed values very different to that exhibited by the bulk Nb, because of the low thickness of these materials. Moreover, a change in the Hall coefficient signs from negative to positive and vice-versa was observed, attributed to several effects, based on the presence of two type of carriers, holes and electrons; the positive value of RH exhibited by thin films with low nitrogen concertation was attributed to this small quantity of nitrogen atoms mainly dispersing the electrons, favoring the hole type carries mobility. As the nitrogen levels increased, the nitrogen atoms were no longer impurities and the material behaved as an Nb thin film.

**Author Contributions:** Conceptualization, A.G.-F., H.A.C. and W.d.l.C.; methodology, A.G.-F.; validation, H.A.C., E.R.-P., and A.G.-F.; formal analysis, H.A.C. and E.R.-P.; investigation, A.G.-F., W.d.l.C., and H.A.C.; resources, W.d.l.C.; data curation, A.G.-F., W.d.l.C., E.R.-P., and H.A.C.; writing—original draft preparation, A.G.-F., H.A.C., and E.R.-P.; writing—review and editing, A.G.-F., H.A.C., W.d.l.C., E.R.-P., and D.E.-R.; visualization, E.R.-P., and H.A.C.; supervision, W.d.l.C.; project administration, W.d.l.C.; funding acquisition, W.d.l.C.

**Funding:** This work was partially supported by DGAPA-UNAM IN112918 grant. A.G.-F. received a scholarship from CONACyT-México.

**Acknowledgments:** The authors are grateful to Eduardo Murillo, Israel Gradilla, and Jesús Antonio Díaz for valuable technical assistance.

**Conflicts of Interest:** The authors declare no conflict of interest.

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
