# Peer review of "Correlation Between Stoichiometry of NbxNy Coatings Produced by DC Magnetron Sputtering with Electrical Conductivity and the Hall Coefficient"

_coatings, doi:10.3390/coatings9030196_

Round 1
Reviewer 1 Report
-Authors could rewritte the first part of abstract in order to avoid the « basic » results. The second part of this abstract is relevent. For instance, authors use XPS to highlight Nb–N bonds. This is an expected "basic" result
Introduction :
Authors should more develop their motivations : It seems to be original to correlate electric transport properties and deposition parameters (nitrogen content in the gas mixture) but what kind of advantage is expected from these results.
-Line 75 : 10-5 instead of 10-5 (same for N2)
-Line 79 : W instead of watts ?
-Line 82 : Authors could use sccm instead of ml/min, this could be changed everywhere
-Lines 114 – 119 : I do not think that the intrinsic sputtering yield of Nb is changed. Obviously I agree with authors about poisonning of the Nb target by nitrogen, which leads to an increase of the nitrogen content in the film.
The references are not relevent concerning the reactive mode of the magnetron sputtering process (see the papers from Berg or Billard).
-Lines 145-153 : It is difficult to understand why authors give nitrogen contents measured by both XPS and AES analyses. Authors explain the differences between these results. this part seems to be "off topic".
-Line 157 : What kind of application ?
Is it possible to perform XRD analysis of these films because, obviously, structure
and microstructure are important regarding electrical conductivity and
hall coefficient?
Author Response
Attached, we are sending the response to reviewer 1

Reviewer 2 Report
The paper reports the results concerning the effect of nitrogen flow on the elemental composition and electrical properties of magnetron sputtered NbN films. From my point of view, the manuscript needs substantial revision because it gives rise to many questions remaining unanswered and some of the presented results are not justified. However, the paper could be suitable for publication after the following comments are addressed by the authors.
1. Section 1 (Introduction) presents an adequate overview of the work done earlier. However, the motivation of the study of room-temperature electrical properties of NbN films is not adequately justified.
2. Section 2 (Materials and Methods). It is unclear how the film thickness was determined.
3. It is reported in Section 2 that the N2 flow was varied between 0 and 4 mL/min. However, the results in Section 3 are presented only for the N2 flow from 0 to 2.5 mL/min.
4. Page 3, line 106. “This behavior agrees with the results of the stoichiometry based on the XPS measurements…”. Since both the results are based on the same XPS measurements, it would be surprising disagreement between them. Therefore, this phrase should be redrafted.
5. Page 4, lines 112-115. “This dependence is expected because the increase in the nitrogen flow rate will decrease the sputtering yield of the Nb atoms due to the formation of a rich nitrogen surface layer on the target surface by the bombardment of argon and nitrogen ions, which is a very common process known as poisoning of the target [27,28]”. Poisoning of the target should substantially decrease the sputtering yield, because for nitrides it is several times lower than that of pure metals (see e.g. Rauch et al. Surf. Coat. Technol. – 2002. – V. 157. – P. 138-142). This results in a substantial decrease in the deposition rate and, consequently, in the film thickness, whereas the authors report that the thickness of the films studied was slightly changed. Therefore, the most likely reason for the increase in nitrogen content in the films with increasing its flow could be that the films deposited at low N2 flows were nitrogen-deficient due to too low N2 partial pressure. On the other hand, the films obtained at higher N2 flows and partial pressures tend towards the stoichiometric composition.
6. The previous comment also concerns the following phrase: “a lower deposition rate indicates that fewer metallic atoms reach the substrate, thereby increasing the N/Nb ratio [29] (page 4, lines 116-117). This statement is in disagreement with the film thickness data reported in Section 2.
7. Page 4, line 119. “…a shift is also observed for the peaks of both elements”. There are no any peaks in Fig. 2. Therefore, this phrase should be redrafted. The same is true for the sentence given in lines 124-125.
8. Page 4, line 124. “The shifting of Nb 3d peak is observed in a range between 202.5 eV and 205.5 eV…”. The binding energy of Nb shown in Fig. 2 varies from ~202.5 to ~205.0 eV.
9. Page 4, lines 126-128. “This shifting is due to the changes in the stoichiometry and also due to the possible formation of different phases, as the nitrogen concentration is increased”. What phases are meant?
10. It is not clear why two different methods of determination of elemental composition of the films were used. Do these methods duplicate or supplement each other? Anyway, experimental errors should be provided in Tables 1 and 2. In addition, taking into account that the experimental error in determination of elemental composition usually reaches 1-2 %, it is reasonable to round the presented values at least to one decimal point.
11. It is more appropriate to represent film composition in terms of atomic percentage as Nb1-xNx rather than NbNx.
12. The film composition at a N2 flow of 1.0 mL/min is given as NbN0.39 (Table 1), NbN0.32 (Table 2) and NbN0.12 (Table 3). Why in the latter case the nitrogen percentage is so low? This film is not mentioned in section 3.1, why it was chosen for investigation of electrical properties?
13. Page 6, lines 179-180. “…which is often observed in this kind of nanocrystalline tending to amorphous thin films…”. This phrase is very speculative, because there are absent any results of the investigation of film microstructure in the manuscript.
14. Page 5, line 155. “…we decided to study the region around room temperature…”. In fact, the authors report the results obtained in the temperature range 85-350 K. It seems the deviations from room temperature are too much. What temperature the authors used as a room temperature. This value should be clearly indicated at least when considering the results presented in Table 3.
15. Page 7, lines 203-204. “…(i) Nb band structure in amorphous thin films…”. What is Nb band structure? This band structure and amorphous structure of the films are not justified by the results reported in the manuscript.
Page 7, lines 215-216. “…because of the changes in the band structure, due to the formation of a SiNbN interlayer between the substrate and the coating…”. The formation of the SiNbN interlayer is not also justified by the results.
Author Response
Attached, we are sending the response to reviewer 2

Reviewer 3 Report
The article reports on NbNx coatings produced by dc magnetron sputtering with different stoichiometry and their electrical conductivity and Hall coefficient. Prior to a publication, the paper should be completed and modified according to the following comments.
Comments and recommendation
1. It is worth to mention electrocatalytic properties of NbC in the introduction. See https://doi.org/10.1021/acsami.7b10317
2. Structure and phase composition of nitrides can vary significantly under different nitrogen flow ratio. See https://doi.org/10.1016/j.apsusc.2018.06.129
3. The authors state in the introduction that grain boundaries have significant influence on conductivity. However, there is no information on crystal structure and grain size of the coatings. Therefore, I strongly recommend to provide XRD measurements for the coatings that must allow to analyze how nitrogen content influences the structural state of the coatings and so conductivity.
4. XPS. In line 126-128 authors explain the peak shift. However, the provided interpretation is not complete. The XPS peak shift for metallic and non-metallic elements is connected with charge transfer, which differs for these atoms. Please, refer to a handbook on XPS (e.g. Siegfried Hofmann, Auger- and X-Ray Photoelectron Spectroscopy in Materials Science) or articles (e.g. Surface and Coatings Technology 151–152 (2002) 194–203, Journal of Electron Spectroscopy and Related Phenomena 135 (2004) 27–39, Surface Science Reports 6 (1987) 253-415), and give more deeper interpretation of you results that will also provide you with more insights on conductivity.
5. The statement “This behavior is ascribed to the great quantity of grain boundaries and defects scattering effect, that is a typical characteristic of amorphous materials” through the text is physically incorrect! Amorphous material has no grains! Please, clarify!
The article contains very valuable experimental outputs on NbNx coatings produced by dc magnetron sputtering with different stoichiometry and their electrical conductivity and Hall coefficient, but they should be analyzed more detailed and interpreted correctly. Therefore, the major revision is necessary.
Author Response
Attached, we are sending the response to reviewer 3

Round 2
Reviewer 1 Report
Thank you for your anwsers.
The details about target poisonning as well as structure analysis could be improved.
Nevertheless, arguments appear sufficient to comment the center points of this paper.
Best regards
Author Response
Thank you for your comments
Corrections were done
Reviewer 2 Report
The revised manuscript provides clearer presentation and justification of the results obtained. However, there are still some critical points which should be improved before publishing the paper.
1. Abbreviation “RT” is explained long after (line 204) its appearance in the text (line 69).
2. Comment 10 was not satisfactorily addressed by the authors. First, experimental errors should be provided in Tables 1 and 2. Second, taking into account that the experimental error in determination of elemental composition usually reaches 1-2 %, it is reasonable to round the presented values at least to one decimal point.
3. In the response to Comment 12 the authors claim that “In all text NbN0.12 was replaced by NbN0.32 and the reason of the chosen of these stoichiometries is given in the Auger section”. However, NbN0.12 film is still shown in Fig. 4.
4. It is reasonable to replace “Nb1.0N1.0“ by “NbN” (line 199, etc.).
5. Lines 201-202: “…which is often observed in this kind of nanocrystalline tending to amorphous thin films…”. The words “this kind of” should be removed because the authors do not exactly know the microstructure of their films.
6. Grammar and phrasing mistakes are still present in the revised manuscript, therefore, it is in need of an in-depth English review.
Author Response
Dear Reviewer
Thank you for your corrections
Reviewer 3 Report
The authors have improved the manuscript according to my comments and suggestions. However, some points have not been fulfilled:
The presence of the broad peak in XRD diffractogram indicates the absence of any long-range order, and therefore can be a fingerprint of amorphous and also nanocrystalline materials. In an amorphous material, a broad peak reflects the first shell neighbour distance of the constituent atoms of clusters, while in a nanocrystalline material is the nanograin size. The positions of the broad XRD peak may also change with N2 flow and thus indicate that there are changes in short-range ordering of the amorphous material. Therefore, it is very important to provide all the XRD diffractograms and their analysis! Please, consider this for analysis and calculate claster/nanograin size from XRD if possible.
Authors state: "Considering that the “grain boundaries” concept is not adequate; this expression was changed by: “highly deformed regions”. For disordered materials, it is important to note that the density of deformed zones can be considered high." It is not adequate to use “highly deformed regions” phrase instead of “grain boundaries”. Generally, it is commonly accepted that amorphous materials composed of short-range ordered clusters. Spatial order in the positioning of atoms in a material with amorphous structure has topological nature while the ordering of different types of atoms has chemical or compositional nature.
It is important to provide the interpretation on why the NbN films grow amorphous but not crystalline.
Author Response
Thank you for your comments

Round 3
Reviewer 3 Report
The manuscript has been improved by the authors following my suggestion and comments. I recommend to publish the manuscript in the journal Coatings.